# Learning New Tricks From Old Dogs: Multi-Source Transfer Learning From Pre-Trained Networks

**Joshua Ka-Wing Lee**
Dept. EECS, MIT
jk_lee@mit.edu

**Prasanna Sattigeri**
MIT-IBM Watson AI Lab, IBM Research
psattig@us.ibm.com

**Gregory W. Wornell**
Dept. EECS, MIT
gww@mit.edu

## Abstract

The advent of deep learning algorithms for mobile devices and sensors has led to a dramatic expansion in the availability and number of systems trained on a wide range of machine learning tasks, creating a host of opportunities and challenges in the realm of transfer learning. Currently, most transfer learning methods require some kind of control over the systems learned, either by enforcing constraints during the source training, or through the use of a joint optimization objective between tasks that requires all data be co-located for training. However, for practical, privacy, or other reasons, in a variety of applications we may have no control over the individual source task training, nor access to source training samples. Instead we only have access to features pre-trained on such data as the output of "black-boxes." For such scenarios, we consider the multi-source learning problem of training a classifier using an ensemble of pre-trained neural networks for a set of classes that have not been observed by any of the source networks, and for which we have very few training samples. We show that by using these distributed networks as feature extractors, we can train an effective classifier in a computationally-efficient manner using tools from (nonlinear) maximal correlation analysis. In particular, we develop a method we refer to as maximal correlation weighting (MCW) to build the required target classifier from an appropriate weighting of the feature functions from the source networks. We illustrate the effectiveness of the resulting classifier on datasets derived from the CIFAR-100, Stanford Dogs, and Tiny ImageNet datasets, and, in addition, use the methodology to characterize the relative value of different source tasks in learning a target task.

## 1 Introduction

Recently, the development of efficient algorithms for training deep neural networks on diverse platforms with limited interaction has created both opportunities and challenges for deep learning. An emerging example involves training networks on mobile devices [8, 23, 14]. In such cases, while each user's device may be training on a different set of data with a different classification objective, multi-task learning techniques can be used to leverage these separate datasets in order to transfer to new tasks for which we observe few samples.

However, most existing methods require some aspect of control over the training on the source datasets. Either all the datasets must be located on the same device for training based on some joint optimization criterion, or the overall architecture requires some level of control over the training for each individual source dataset. In the case of, e.g., object classification in images collected by users, sending this data to a central location for processing may be impractical, or even a violation of privacy rights. Alternatively, it is possible that one might wish to use older, pre-trained classifiers for which the original training data is no longer available, and to transfer them for use in a new task. In either case, it could be acceptable to transmit the neural network features learned by the device in an

anonymized fashion, and to then combine the networks learned by multiple users in order to classify novel images.

This would be an example of a multi-task learning problem in which we have not only multiple source datasets, but access to only pre-trained networks (whose learning objective we cannot control) from those datasets, not the underlying training data used, and we wish to train a classifier for some new target label set given only a few target samples.

Fine-tuning methods can be used when the source network is frozen to transfer to a target domain, but these methods tend not to work very well in a few-shot setting when there are multiple networks due to the number of parameters necessary for fine-tuning, especially in an environment where features cannot be learned with the intention of transfer [4].

In this paper, we apply the methodology of (nonlinear) maximal correlation analysis that originated with Hirschfeld [9] to this problem. In particular, we exploit a useful and convenient interpretation of the features in a neural network as maximal correlation functions, as described in, e.g., [10]. The result is a method we refer to as maximal correlation weighting (MCW) for combining multiple pre-trained neural networks to carry out few-shot learning of a classifier to distinguish a set of never-before-seen classes. Attractively, this method allows for the computation of combining weights on individual feature functions in a completely decoupled fashion.

This paper is organized as follows. In Section 2, we describe the problem formulation and related work. In Section 3, we introduce the the relevant aspects of maximal correlation analysis for combining neural networks, and develop the MCW methodology and processing used to train our classifier. Section 4 describes experimental results on the CIFAR-100, Stanford Dogs, and Tiny ImageNet datasets, and Section 5 contains concluding remarks.

## 2 Background and Problem Description

### 2.1 Problem Formulation and Notation

Consider a multi-task learning setup in which we have $N$ different source classification tasks $\{\mathcal{T}_1^s, \ldots, \mathcal{T}_N^s\}$, for which we have labeled data $\{(x_1^{s_n}, y_1^{s_n}), \ldots, (x_{k_n}^{s_n}, y_{k_n}^{s_n})\}$ for task $\mathcal{T}_n^s$, $n \in \{1, \ldots, N\}$. We also have a single target task $\mathcal{T}^t$, with associated labeled data $\{(x_1^t, y_1^t), \ldots, (x_k^t, y_k^t)\}$.

For this problem we assume that $x_i^{s_n} \in \mathcal{X}$ for all $n$ and $i$, and $x_i^t \in \mathcal{X}$ for all $i$. That is, the data for the target and each source task are drawn from some common alphabet (e.g., all data are natural images). We do not assume any overlap between labels for any pair of datasets (i.e., $y_i^{s_n} \in \mathcal{Y}^{s_n}$ for all $n$ and $i$, and $y_i^t \in \mathcal{Y}^t$ for all $i$, where $\mathcal{Y}^t \neq \mathcal{Y}^{s_1} \neq \cdots \neq \mathcal{Y}^{s_N}$).

For each source task $\mathcal{T}_n^s$, we have access to a pre-trained neural network which we assume to have been trained to classify $y^{s_n}$ from $x^{s_n}$. We assume that the network has some number of layers corresponding to the extraction of features from $x^{s_n}$, followed by a final classification layer which maps the features to a predicted class label $\hat{y}^{s_n}$. We denote the output of the penultimate layer as $\mathbf{f}^{s_n} : \mathcal{X} \to \mathbb{R}^{l_n}$, of which the $i$th feature is $f_i^{s_n} : \mathcal{X} \to \mathbb{R}$, where $l_n$ is the number of features output by this layer. We denote the final layer as $h^{s_n} : \mathbb{R}^{l_n} \to \mathcal{Y}^{s_n}$, so that the entire neural network classifier can be written as $\hat{y} = (h^{s_n} \circ \mathbf{f}^{s_n})(x)$.

We seek to train a classifier on the target task given training samples $\{(x_1^t, y_1^t), \ldots, (x_k^t, y_k^t)\}$, with access to $h^{s_n}$ and $\mathbf{f}^{s_n}$ for each source dataset, but without any access to the underlying source training samples $\{(x_1^{s_n}, y_1^{s_n}), \ldots, (x_{k_n}^{s_n}, y_{k_n}^{s_n})\}$.

As an example context, this reflects a situation in which there are many devices collecting and analyzing data, but where the target learner is not allowed to access the data, either because the devices have limited bandwidth and cannot transmit everything they have detected, the data is personal (i.e. pictures taken by users of a mobile app) and cannot be transmitted for privacy purposes, or the original data is otherwise lost (if the data was collected a long time ago). However, in these cases, it may still be possible to query the classifier trained on each device to get their intermediate features, which would require less information to be transmitted.

## 2.2 Related Work

Multi-task learning is a well-studied problem, with several variations and formulations. One standard approach is to learn a common feature function $f(\cdot)$ across all tasks which optimize some joint objective, followed by a final classification layer for each task [19, 24]. This is a technique which has some theoretical guarantees as given by Ben-David, et al., [2]. While effective, this method requires joint training, which our formulation precludes.

Gupta and Ratinov [7] propose a method of combining the outputs of multiple pre-trained classifiers by training on their raw predictions, but this method is designed for pre-trained classifiers specially selected to work well in combination with the target task, with an emphasis on cases where the number of possible class labels (i.e. the value of each $|\mathcal{Y}^{s_n}|$) is large, which we do not assume in our problem formulation.

Other methods involve some kind of sequential learning [27] or shared memory unit [18], which could decentralize data storage, but which still require joint control over the training [17].

Meta-learning algorithms have also gained popularity in recent years [21, 22]. These algorithms attempt to learn a suitably general learning rule or model from a set of source tasks which can be fine-tuned with data from a target task [4]. While these methods allow for the combining of multiple source datasets, they are still bound by the need for centralized training.

Finally, the notion of transferring from a single pre-trained network onto a new target task has also been studied before. Yosinski, et al., explore the transferability of different layers of a neural net to other tasks in the context of learning general features [28], while Bao, et al., propose a score for measuring transferability of features across tasks [1].

## 3 Multi-Source Transfer Learning via Maximal Correlations

### 3.1 Maximal Correlation Analysis

Our methodology is based on the use of maximal correlation analysis, which originated with the work of Hirschfeld [9], and has been further developed in a wide range of subsequent work , including by Gebelein and Rényi [6, 26], and as a result is sometimes referred to as Hirschfeld-Gebelein-Renyi (HGR) maximal correlation analysis. (For a more detailed summary of this literature, see, e.g., the references and discussion in [10].)

Given $1 \leq k \leq K - 1$ with $K = \min\{|\mathcal{X}|, |\mathcal{Y}|\}$, the maximal correlation problem for random variables $X \in \mathcal{X}$ and $Y \in \mathcal{Y}$ is

$$(\mathbf{f}^*, \mathbf{g}^*) \triangleq \underset{\substack{\mathbf{f}\colon \mathcal{X}\to\mathbb{R}^k,\ \mathbf{g}\colon \mathcal{Y}\to\mathbb{R}^k \\ \mathbb{E}[\mathbf{f}(X)]=\mathbb{E}[\mathbf{g}(Y)]=\mathbf{0}, \\ \mathbb{E}[\mathbf{f}(X)\mathbf{f}^{\mathrm{T}}(X)]=\mathbb{E}[\mathbf{g}(Y)\mathbf{g}^{\mathrm{T}}(Y)]=\mathbf{I}}}{\arg\max} \mathbb{E}\left[\mathbf{f}^{\mathrm{T}}(X)\,\mathbf{g}(Y)\right], \tag{1}$$

where expectations are with respect to joint distribution $P_{X,Y}$. We refer to $\mathbf{f}^*$ and $\mathbf{g}^*$ as the maximal correlation functions. With $\mathbf{f}^* = (f_1^*, \ldots, f_k^*)^{\mathrm{T}}$ and $\mathbf{g} = (g_1^*, \ldots, g_k^*)^{\mathrm{T}}$, we further define the associated maximal correlations $\sigma_i = \mathbb{E}\left[f_i^*(X)\,g_i^*(Y)\right]$ for $i = 1, \ldots, k$. In turn, the optimizing functions satisfy

$$\mathbb{E}_{P_{X|Y}(\cdot|y)}\left[f_i^*(X)\right] = \sigma_i\,g_i^*(y) \quad \text{and} \quad \mathbb{E}_{P_{Y|X}(\cdot|x)}\left[g_i^*(Y)\right] = \sigma_i\,f_i^*(y),$$

which underlies the alternating conditional expectations (ACE) algorithm of Breiman and Friedman [3] for computing these functions. Indeed, for a given $\mathbf{f}$, the correlation maximizing $\mathbf{g}$ has components

$$\hat{g}_i(y) \propto \mathbb{E}_{P_{X|Y}(\cdot|y)}\left[f_i^*(X)\right], \quad i = 1, \ldots, k. \tag{2}$$

As described in [11, 10], the maximal correlation problem is a variational form of a modal decomposition (i.e., generalized SVD) of joint distributions of the form

$$P_{X,Y}(x,y) = P_X(x)\,P_Y(y)\left[1 + \sum_{i=1}^{K-1} \sigma_i\,f_i^*(x)\,g_i^*(y)\right], \tag{3}$$

via which predictions are made according to

$$P_{Y|X}(y|x) = P_Y(y) \left( 1 + \sum_{i=1}^{k} \sigma_i f_i^*(x) g_i^*(y) \right),$$ (4)

where suitable estimates of $P_Y$ are obtained from the data or domain knowledge about label distributions.

Moreover, the maximal correlation features arise naturally in a local version of softmax regression [10], and thus have a direct interpretation in the context of neural networks. In particular, given (normalized) features $\mathbf{f}$, [10] shows that such regression produces $\mathbb{E}_{p_{X|Y}(\cdot|y)}[\mathbf{f}(X)]$ as combining weights. Moreover, [10] establishes that optimizing over the choice of features yields the maximal correlation ones, i.e., $\mathbf{f}^*$, and that as a result the corresponding combining weights correspond to $\mathbf{g}^*$ (weighted by $\sigma_1, \ldots, \sigma_k$). (And as as such, it also highlights the connection between the ACE algorithm and the use of traditional neural network training.)

### 3.2 Combining Maximal Correlation Functions

The preceding relationships motivate our approach to their application to the multi-task learning problem. Given a fixed set of feature functions $\{\mathbf{f}^{s_1}, \ldots, \mathbf{f}^{s_N}\}$ we seek to maximize the total maximal correlation

$$\mathcal{L} = \mathbb{E}_{\hat{P}_{X,Y}^t} \left[ \mathbf{f}^{\mathrm{T}}(X) \, \mathbf{g}(Y) \right]$$ (5)

with respect to $\mathbf{g}$, where $\mathbf{f} = (\mathbf{f}^{s_1}, \ldots, \mathbf{f}^{s_N})^{\mathrm{T}}$ and $\mathbf{g} = (\mathbf{g}^{s_1}, \ldots, \mathbf{g}^{s_N})^{\mathrm{T}}$, and where the optimization is over all valid (zero-mean and unit-variance with respect to the empirical distribution of the target class labels) $\mathbf{g}$ for fixed $\mathbf{f}$. $\hat{P}_{X,Y}^t$ is the empirical joint target distribution of $X$ and $Y$.

Expanding (5) as

$$\mathcal{L} = \sum_{i,n} \mathbb{E}_{\hat{P}_{X,Y}^t} \left[ f_i^{s_n}(X) \, g_i^{s_n}(Y) \right],$$ (6)

we can then maximize each term separately, yielding

$$g_i^{s_n}(y) = \arg\max_{\tilde{g}_i^{s_n}} \mathcal{L} = \arg\max_{\tilde{g}_i^{s_n}} \mathbb{E}_{\hat{P}_{X,Y}^t} \left[ f_i^{s_n}(X) \tilde{g}_i^{s_n}(Y) \right].$$ (7)

Then, for each $g_i^{s_n}(y)$, for a fixed $f_i^{s_n}$, we have from (2) that the optimal $g_i^{s_n}$ is given by the conditional expectation

$$g_i^{s_n}(y) = \mathbb{E}_{\hat{P}_{X|Y}^t(\cdot|y)} \left[ f_i^{s_n}(X) \right],$$ (8)

which can easily be computed from the target samples.

In turn, we compute the corresponding maximized correlation for each pair of functions $f_i^{s_n}$ and $g_i^{s_n}$ via

$$\sigma_{n,i} = \mathbb{E}_{\hat{P}_{X,Y}^t} \left[ f_i^{s_n}(X) \, g_i^{s_n}(Y) \right].$$ (9)

### 3.3 The Maximal Correlation Weighting (MCW) Algorithm

Using the combining weights thus derived, a predictor for the target labels is formed in accordance with (4); specifically,

$$\hat{P}_{Y|X}(y|x) = \hat{P}_Y^t(y) \left( 1 + \sum_{n,i} \sigma_{n,i} f_i^{s_n}(x) g_i^{s_n}(y) \right),$$ (10)

from which are classification $y$ for a given test sample $x$ is

$$\hat{y} = \arg\max_y \hat{P}_{Y|X}(y|x) = \arg\max_y \hat{P}_Y^t(y) \left( 1 + \sum_{n,i} \sigma_{n,i} f_i^{s_n}(x) g_i^{s_n}(y) \right),$$ (11)

where $\hat{P}_Y^t$ is an estimate of the target label distribution.

---

**Algorithm 1** Extracting maximal correlation parameters

---

**Data:** zero-mean, unit-variance feature functions $\{f_i^{s_n}\}$ from source tasks and target task samples
$\qquad \{(x_1^t, y_1^t), \ldots, (x_k^t, y_k^t)\}$
**Result:** associated maximal correlations $\{\sigma_{n,i}\}$ and correlation functions $\{g_i^{s_n}\}$
**for** $n = 1, \ldots, N$ **do** // Iterate over all source tasks
$\quad$ **for** $i = 1, \ldots, l_n$ **do** // Iterate over features in each network
$\qquad$ **for** $y \in \mathcal{Y}^t$ **do** // Iterate over all target class labels
$\qquad\quad g_i^{s_n}(y) \quad\leftarrow\quad \mathbb{E}_{P_{X|Y}^t(\cdot|y)}\left[f_i^{s_n}(X)\right]$ // Compute feature and label-specific
$\qquad\quad$ weight
$\qquad$ **end**
$\qquad \sigma_{n,i} \leftarrow \mathbb{E}_{\hat{P}_{X,Y}^t}\left[f_i^{s_n}(X)\, g_i^{s_n}(Y)\right]$ // Compute feature-specific weight
$\quad$ **end**
**end**
**return** $\{g_i^{s_n}\}, \{\sigma_{n,i}\}$

---

---

**Algorithm 2** Prediction with the maximal correlation weighting method

---

**Data:** maximal correlation functions $\{f_i^{s_n}\}$ and $\{g_i^{s_n}\}$ with associated correlations $\{\sigma_{n,i}\}$, empirical
$\qquad$ class label distribution $\hat{P}_Y^t$, and target task sample $x^t$
**Result:** class label prediction $\hat{y}^t$ given $x^t$
Initialize $\hat{P}_{Y|X}^t(y|x^t) = \hat{P}_Y^t(y)\ \forall y \in \mathcal{Y}^t$
**for** $n = 1, \ldots, N$ **do** // Iterate over all source tasks
$\quad$ **for** $i = 1, \ldots, l_n$ **do** // Iterate over features in each network
$\qquad$ **for** $y \in \mathcal{Y}^t$ **do** // Iterate over all target class labels
$\qquad\quad \hat{P}_{Y|X}^t(y|x^t) = \hat{P}_{Y|X}^t(y|x^t) + \hat{P}_Y^t(y)\sigma_{n,i}f_i^{s_n}(x^t)g_i^{s_n}(y)$ // Apply Equation (9)
$\qquad$ **end**
$\quad$ **end**
**end**
**return** $\arg\max_y \hat{P}_{Y|X}^t(y|x)$

---

The resulting algorithms for learning the MCW parameters and computing the MCW predictions are summarized in Algorithm 1 and Algorithm 2.

Computing the empirical conditional expected value requires a single pass through the data, and so has linear time complexity in the number of target samples. We also need to compute one conditional expectation for each feature function. Thus, the time complexity of the fine-tuning is $O(C + NKk)$, where $C$ is the time needed to extract features from all the pre-trained networks, $N$ is the number of networks, $K$ is the maximum number of features per network, and $k$ is the number of target training samples. The number of parameters grows as $O(NK|\mathcal{Y}^t|)$, which is the number of entries needed to store all the $g$ functions. $|\mathcal{Y}^t|$ is the number of target class labels.

To compute a prediction from one target test sample, the time complexity is $O(C + NK|\mathcal{Y}^t|)$. This arises from the fact that we must compute the quantity $\sum_{n,i} \sigma_{n,i} f_i^{s_n}(x) g_i^{s_n}(y)$ for each possible class label.

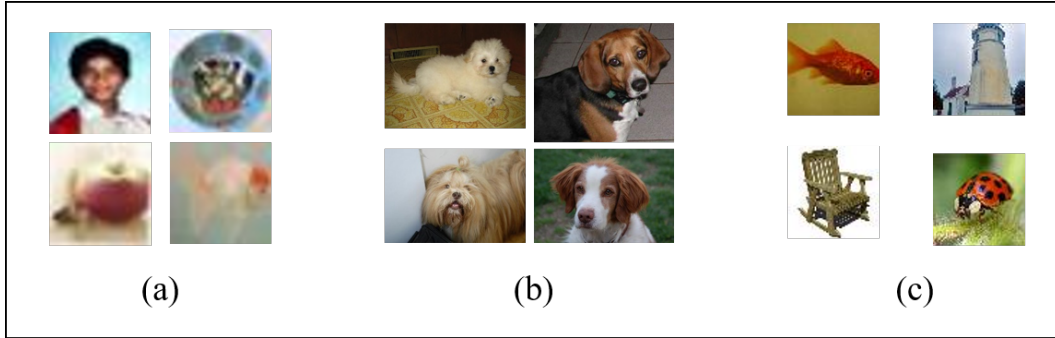

Figure 1: Example images from the (a) CIFAR-100, (b) Stanford Dogs, and (c) Tiny ImageNet datasets.

## 4  Experimental Results

### 4.1  General Experimental Setup

In order to illustrate the effectiveness of the MCW method, we perform experiments on three different image classification datasets: CIFAR-100, Stanford Dogs, and Tiny ImageNet. Example images from each dataset can be found in Figure 1.

For each dataset, we divide the classes into a set of mutually exclusive subsets, select one subset as our target task, and several others as the source datasets. We use the LeNet architecture [15] as our neural network for each source dataset, and train a different network for each source dataset. We implemented the network in PyTorch [25], and trained it with learning rate=0.001, momentum=0.9, and number of epochs = 100.

We remove the means and normalize to unit variance all of the feature functions with respect to the target samples, and then compute the maximal correlations and associated functions for each output in the penultimate layer using the target data according to Algorithm 1. We then use them to compute predictions on the test set for the target task according to Algorithm 2.

We compare the classification accuracies on the test set with that of a Support Vector Machine (SVM) trained on the penultimate layers with the same target training data (similar to the setup in [7]), as well as the best results from the MCW method and SVM method using only one source dataset/neural network. We also include the "upper bound" baseline performance on the dataset by a LeNet neural network trained on a number of target training samples equal to the number of training samples provided for each source dataset. The reported results are over 20 runs using the same set of tasks for each run.[1]

### 4.2  CIFAR-100 Dataset

The CIFAR-100 dataset[2] [13] is a collection of color images of size 32x32 drawn from 100 different categories of real-world subjects. Because of the low resolution of the images, CIFAR-100 is generally seen as a difficult classification problem. For our experiment, we construct a series of binary classification tasks from the classes. We randomly selected "apple" vs. "fish" as our target binary classification task, and randomly selected 10 other pairs of non-overlapping categories for the source tasks. For each source task, we extracted 500 samples per class for training, and we used 1, 5, 10, and 20 samples per class to compute the maximal correlation functions in the target task. We used the training/test splits included with the dataset, and report results over all test samples with the target labels.

Table 1 shows the test accuracies of our algorithm as applied to the CIFAR-100 dataset. We can see that the MCW method performs significantly better than an SVM when there are few samples,

Table 1: Experimental results for the CIFAR-100 dataset. Accuracies are reported with 95% confidence intervals.

| Method | 1-Shot Acc. | 5-Shot Acc. | 10-Shot Acc. | 20-Shot Acc. |
|---|---|---|---|---|
| Best Single Source SVM | $56.9 \pm 2.5$ | $67.0 \pm 3.0$ | $70.4 \pm 1.9$ | $70.9 \pm 1.2$ |
| Best Single Source MCW | $59.2 \pm 2.1$ | $69.0 \pm 3.0$ | $67.0 \pm 2.4$ | $70.4 \pm 1.5$ |
| Multi-Source SVM | $64.7 \pm 3.0$ | $72.8 \pm 2.7$ | $76.2 \pm 1.8$ | $81.5 \pm 0.6$ |
| **Multi-Source MCW** | $\mathbf{69.0 \pm 3.0}$ | $\mathbf{78.1 \pm 0.8}$ | $\mathbf{80.1 \pm 0.8}$ | $\mathbf{81.7 \pm 0.6}$ |
| Baseline (All Target Samples) | | $90.7 \pm 0.1$ | | |

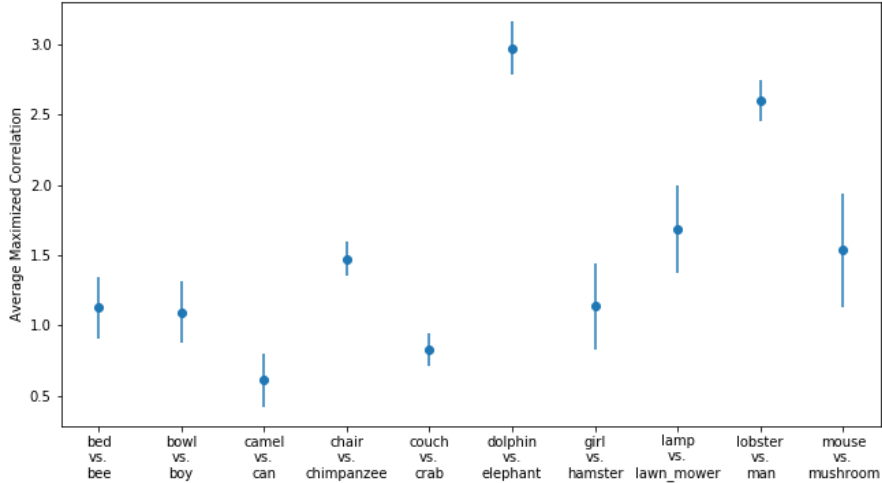

Figure 2: Average values of $\sum_i \sigma_{n,i}$ for each source task $s_n$ for the 5-shot transfer learning task on the CIFAR-100 dataset, with the target task of "apple vs. fish." Points are plotted with 95% confidence intervals.

likely due to its ability to work with fewer target data points in learning, but that this performance gap closes as more target training samples are added, likely due to the fact that the models which require joint training over the features begin to have enough target samples to properly learn their parameters. In addition, we can see that combining multiple networks provides performance that is better than any one network can achieve with the same methods, once again suggesting that our algorithm is taking in contributions from multiple sources instead of just one.

In order to investigate the functioning of the MCW method, we plot the sum of correlations for each of the 10 tasks for the 5-shot case in Figure 2. We can see a significant variation among tasks, which provides a clear indication of which tasks are being preferred and which do not contribute as much to the overall performance. To verify this, we run two additional experiments in which we first remove the source task with the lowest total correlation ("camel" vs. "can") and see how well the MCW method performs with the remaining 9 source datasets, and then remove the task with the highest total correlation ("dolphin" vs. "elephant") while keeping the other 9 sources in and run the same test.

Without the least-favoured task, the classification accuracy drops to $76.8 \pm 1.0$, which is not a significant difference from using all 10 source tasks. However, when we remove the most-favoured task, the accuracy plummets to $73.0 \pm 1.3$, which indicates that "dolphin" vs. "elephant" had a significant impact on the quality of the classifier, but that the MCW method still takes the input of the other tasks into account in order to construct a good classifier on the target set.

Table 2: Experimental results for the Stanford Dogs dataset. Accuracies are reported with 95% confidence intervals.

| Method | 5-Shot Accuracy |
|---|---|
| Best Single Source SVM | $35.8 \pm 0.8$ |
| Best Single Source MCW | $38.2 \pm 0.6$ |
| Multi-Source SVM | $38.9 \pm 0.3$ |
| **Multi-Source MCW** | **$41.6 \pm 0.5$** |
| Baseline (All Target Samples) | $55.2 \pm 0.1$ |

Table 3: Experimental results for the Tiny ImageNet dataset. Accuracies are reported with 95% confidence intervals.

| Method | 5-Shot Accuracy |
|---|---|
| Best Single Source SVM | $31.4 \pm 0.9$ |
| Best Single Source MCW | $33.9 \pm 1.0$ |
| Multi-Source SVM | $42.5 \pm 1.4$ |
| **Multi-Source MCW** | **$47.4 \pm 1.1$** |
| Baseline (All Target Samples) | $53.8 \pm 0.1$ |

### 4.3  Stanford Dogs Dataset

The Stanford Dogs dataset[3] [12] is a subset of the ImageNet dataset designed for fine-grained image classification. It consists of 22,000 images of varying sizes covering 120 classes of dog breeds. For this task we construct a random 5-way target classification task (differentiating between "Chihuahua", "Japanese Spaniel", "Maltese Dog", "Pekinese", and "Shih-Tzu") and 10 other random 5-way source classification tasks with no overlapping classes. For the target set, we take 5 samples per class for training and use the rest for testing. For the source sets, we take 100 samples per class for training. All images were resized to size 144x144.

Table 2 shows the test accuracies of our algorithm as applied to the Stanford Dogs dataset. This time, we observe a loose hierarchy whereby the MCW method outperforms the SVM, which in turn outperforms any single source transfer. We can thus conclude that the MCW method is effective in the case of $m$-way learning for $m > 2$, and that we can still leverage multiple networks to get a gain in cases where the classes are very similar.

### 4.4  Tiny ImageNet Dataset

The Tiny ImageNet dataset[4] [16] is another subset of the ImageNet dataset, consisting of images of size 64x64 drawn from 200 categories, with 500 images provided for each category. The categories cover a much wider range than the Stanford Dogs dataset, including animals, natural and man-made objects, and even abstract concepts (e.g. "elongation"). As with the Stanford Dogs dataset, we constructed 11 random 5-way classification tasks, and selected one as the target task ("Lighthouse" vs. "Rocking Chair" vs. "Bannister" vs. "Jellyfish" vs. "Chain") and the others as source tasks. We used 5 training samples per class for the target task (with 250 samples per class for testing) and all 500 samples per class for the source training samples. For the baseline, we only trained with the 250 samples per class in the target dataset that were not in the test split.

Table 3 shows the test accuracies of the MCW method as applied to the Tiny ImageNet dataset. Compared to the Stanford Dogs dataset, we see a larger gain from leveraging multiple sources compared to a single source, which suggests that if the source classes are much more dissimilar than the target classes, then integrating more networks (and thus leveraging a wider range of features) will have a greater effect on target task accuracy, likely due to the ability of different source tasks to "cover" the feature set needed for the target task, as opposed to the Dogs setup where the classes were highly similar.

# 5 Concluding Remarks

We presented a new multi-task learning problem inspired by advances in the modern Deep Learning ecosystem in which a target task learner has access to only a few target task samples, and access to the neural networks already trained by the sources, but not the underlying data. By leveraging the Hirschfeld-Gebelein-Rényi maximal correlation, we were able to develop a fast, easily-computed method for combining the features extracted by these neural networks to build a classifier for the target task.

We showed that this method was effective for binary and 5-way classification on image data, and that combining multiple nets was more effective when there were no similar classes in the source datasets to those in the target dataset.

It is possible that the maximal correlation can also be a tool to measure how important each neural network is relative to training the target task, as we showed in our experiments with the CIFAR-100 dataset. In an online setting, this could encourage a procedure whereby more-relevant networks are queried more often compared to less-relevant networks if data transfer is limited, since it is more important that the more-relevant networks are "correct" (i.e. trained with more training data).

In addition, the privacy implications of our setup could be considered, as it is possible to reconstruct training data from the learned features [5], which means that our method as-is does not erase all privacy concerns. These methods can be countered with differential privacy measures [20], such as adding noise to the feature functions, but their effect on transfer quality is as-of-yet unknown.

Indeed, with the advent of mass small-scale Deep Learning, many opportunities and challenges will arise, allowing us to leverage the power of crowdsourcing for learning in a novel application of the principle of the Wisdom of the Crowd.

### Acknowledgments

This work was supported in part by the MIT-IBM Watson AI Lab, and by NSF under Grant No. CCF-1717610.

## Footnotes

[1]Code for the experiments can be found at http:/allegro.mit.edu/~gww/multitransfer

[2]https://www.cs.toronto.edu/ kriz/cifar.html

[3]http://vision.stanford.edu/aditya86/ImageNetDogs/

[4]https://tiny-imagenet.herokuapp.com/

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
