[Reviews · NeurIPS 2019]

Reviewer 1



This paper presents a novel approach to solve the problem of multi-source transfer learning by using the maximal correlation. The method are evaluated in 3 dataset. However the setting in the three dataset are somehow similar. The dataset are divided into several parts. One is chosen as the target and the rest as sources. The target task and the source tasks are from the same domain in all the experiments. The classifier is also limited to binary or 5-way classifier but not with more ways. The experiment shows that the proposed method has a significant gain in k-shot learning when k <= 10 and it can benefit from multiple sources. It also shows in fig 2 that the maximal correlation is a good metric to show how the new task benefit from a certain old task. The paper is well written and easy to follow.

Reviewer 2



## Summary In the proposed approach, maximal correlations and correlation functions are first obtained from the feature functions and target samples. The maximal correlations and correlation functions are then used to predict the class for the target sample. The evaluation is done on 3 datasets (CIFAR100, Stanford Dogs, and Tiny Imagenet). The proposed MCW method is compared with SVM trained on output of the penultimate layer. For all the datasets, the Multi-Source MCW shows significant advantage especially when there are few samples. ## Originality Formulation of prediction method (MCW) using the maximal correlation and correlation function is novel. The appeal of the method is that no control over the training of the source networks is needed. ## Quality The problem formulation and the method look sound. Source code is not provided with the submission. It would have helped in understanding the method better. Evaluation is a bit weak. There is no comparison with other few-shot learning algorithms. Even though these algorithms might be fundamentally different, ablation studies would have helped in highlighting specific features of the methods. For all the three datasets, only one set of randomly selected set of target classes is chosen. Ideally, the experiments should be carried on multiple subsets of randomly chosen target classes. ## Clarity The problem motivation and introduction is presented well. The problem formulation is also clear. The main section on MCW is a bit dry read. A little introduction to HGR maximal correlation (and correlation functions) and intuitive explanation of the method would be helpful. The algorithm listings, however, are quite helpful. ## Significance The proposed method is simple and elegant. It doesn’t require joint training. It has the potential of opening up new avenues for reusing pre-trained networks. ## Minor Line 110: “We also denote the correlation *of of* the ith …” ## Post rebuttal comments Thank you for the response. Your results on the randomly chosen source and target classes are noted.

Reviewer 3



originality: The paper presents a new method to perform domain adaptation. It originally uses a statistical tool to design a dedicated algorithm. quality and clarity: The method is shown to outperform some alternative methods to perform the task. However, The description of the method lacks (see the specific comments below) both in notation and exposition - it remains unclear why the method performs well, what are the properties of the data that are leveraged, and what makes the method specifically suitable to few-shot learning. significance: The setting addressed is important and seems to become prevalent. Specific comments: From the description in section 3, it is unclear why the method is specifically applicable to low-shot settings. The pseudo-code in Algorithm 1 as presented uses expectation over the underlying P^t_{X|Y} rather than an estimate. line 37 - maybe also mention differential privacy (and state of the art related research and results), as related to the setting. In the related works section, why isn't the method mentioned in lines 91-93 applicable to the paper's setting? should elaborate (also how compares conceptually and performance-wise). Provide reference for the work mentioned in line 101 (Yoshinski) and line 102 (Bao). What is the orthogonality requirement mentioned in lines 115-116 ? line 118: It is stated that the prior P_Y(y) can be estimated from the data. Wouldn't this estimate be very noisy in few-shot settings!? Similarly, in line 129 - the conditional expectation is stated to be easily 'computed', however the correct term is 'estimated' and again, this estimate would be very noisy in few-shot settings.. In formula (2), should make the dependence of g on f explicit. e.g., g_f(y) = .. Some typos: line 110 (of of) line 111 sigma_{i}. Also, shouldn't the f and g be starred at that definition of sigma_{i} !? formula (3): P_{Y|X} formula (9): P_{Y|X} formula (10): P_{Y|X} --------------- Given the detailed answer by the authors and their commitment to address the typos and editorial comments (although a couple of mine remain un-answered) I increase the score to 6

[Author Response · NeurIPS 2019]

We thank the reviewers for their thoughtful insights and comments. Please find below our responses to the main points raised. We have also found a number of typos in addition to the ones listed in the reviews and will correct them in the final version, as well as making improvements based on the comments on readability, clarity, notation, and references.

**1. Testing on randomly chosen source and target classes (R1, R2)**

As suggested in the reviews, in order to strengthen the validity of our results, we reran the experiments for the 5-shot case on the Tiny ImageNet dataset with source and target classes randomly generated each trial and compared MCW and SVM performance. The result was a much wider range of performance ($46.2\pm15.3$ for MCW and $41.7\pm13.9$ for SVM), but the difference in performance between the two methods was consistently in favor of the MCW method ($Accuracy_{MCW} - Accuracy_{SVM} = 4.4 \pm 3.5$). We observe similar phenomena for the other two datasets.

**2. Comparison with other algorithms (R2, R3)**

We first note that the method from 91-93 was designed for use with a large number of target samples, and used an SVM to combine the networks, similar to what we reported with our SVM experiments. In addition, that method required careful selection of "source tasks" to get outputs that could act as features (such as clustering data points in an unsupervised fashion, with the cluster index as the output), which our method does not require.
Experimentally, we also find that applying the method from 91-93 results in performance that is $3.5\pm0.8$ worse than applying the SVM to the penultimate layers of the networks on the Tiny ImageNet setup.
In the future, we intend to perform some ablative studies and compare to other few-shot learning methods which require source-sample access, such as meta-learning techniques.

**3. Revision of Introduction (R2, R3)**

We will revise the introduction to provide a short primer on the HGR Maximal Correlation explaining it as a variational formulation of a particular singular value decomposition of joint distributional information with desirable properties for capturing this information, as well as to add intuition for the MCW method in terms of computing weights in a decoupled fashion for each input feature.

**4. Focus on the few-shot setting (R3)**

Empirically, as shown in Table 1 of the paper, we observed that our method performed best with few samples, with the SVM catching up with our method when we increased the number of samples and eventually overtaking it.
We believe this to be the case because our method operates on each feature independently from the others. As such, it is advantageous in the few-shot setting when there are many features, as our method does not rely on learning ways to combine features. On the other hand, other methods like using an SVM or an additional simple neural net on top of the features requires the learning of a large number of related parameters simultaneously, which is difficult in the few-shot setting as the number of features (and thus parameters) grows large. However, with more samples, the SVM is able to effectively learn how to remove redundant/overlapping features, leading to increased relative performance.

**5. Orthogonality constraint in 115-116 (R3)**

In the definition of HGR Maximal Correlation given in (1), the functions $f_1, ..., f_k$ are constrained to be uncorrelated, as are $g_1, , , , g_k$, but the expression for each optimal $g_i$ given $f_i$ holds without this constraint due to how the objective separates out (as in (5)). For clarity, we will remove the word "orthogonality" and instead say that the constraint that the variables be uncorrelated is unnecessary. Our revised introduction will also explicitly state this constraint.

**6. Few samples resulting in poor estimates for $P_{X|Y}^t$ and $P_Y^t$ from the empirical distributions $\hat{P}_{X|Y}^t$ and $\hat{P}_Y^t$ (R3)**

In general, learning weights and features from few samples is very difficult. However, our method learns the weighting for each feature independently, as opposed to other methods which learn all weights over all features simultaneously. We believe this confers an advantage in the number of samples needed to obtain a good estimate for the weights.
We will also revise the paper to note that marginal distributions can be obtained with unlabeled samples, which are often much easier to collect than labeled samples, so obtaining good estimates of them is comparatively easier.

**7. Justification/Discussion of method performance (R3)**

Based on the feedback from the reviewers, we intend to revise the experimental results/discussion section to include further explanations for why our method performs better or worse in different cases, in addition to the discussion of the choice of the few-shot setting as given in Point 4 of this response.
In particular, for the Stanford Dogs and Tiny ImageNet datasets, we believe that feature redundancy results in less performance gain between the MCW and SVM methods for the Dogs dataset. As a quick experiment, we computed the features for Dogs and Tiny ImageNet again and then, for each feature $f_i$, we looked at the largest correlation it had with the other features in the experiment. Dogs had a larger average correlation coefficient with other features than ImageNet (0.6 vs. 0.49), which suggests a greater amount of redundancy between features for Dogs. Thus, our method results in less gain since the features are more similar and thus adding additional networks adds less additional information which it is able to leverage to outperform the SVM (which has difficulties in dealing with many uncorrelated features in the few-sample regime) and the single-network methods.

[Meta-Review · NeurIPS 2019]

The paper presents a very interesting and efficient technique for transfer learning which is validated on image classification data. The three reviewers agreed on the quality and siginficance of the contributions. I recommend acceptance as a poster presentation.